# IDH2 Inhibitors Gain a Wildcard Status in the Cancer Therapeutics Competition

**DOI:** 10.3390/cancers16193280

**Published:** 2024-09-26

**Authors:** Roberto Piva, Nariman Gharari, Maria Labrador, Sylvie Mader

**Affiliations:** 1Department of Molecular Biotechnology and Health Sciences, University of Turin, 10126 Turin, Italy; nariman.gharari@unito.it (N.G.); maria.labradorgranados@unito.it (M.L.); 2Città della Salute e della Scienza Hospital, 10126 Turin, Italy; 3Department of Biochemistry and Molecular Medicine, Institute for Research in Immunology and Cancer, Université de Montréal, Montreal, QC H3T 1J4, Canada; sylvie.mader@umontreal.ca

**Keywords:** IDH2, breast cancer, immunotherapy, CAR T cells, metabolism, epigenetics

## Abstract

**Simple Summary:**

The studies highlighted in this commentary demonstrate the significant role of wild-type IDH2 in both cancer progression and immune cell function. In triple-negative breast cancer (TNBC), it was found that wild-type IDH2 is essential for tumor survival, as its inhibition leads to disrupted energy metabolism and reduced tumor growth. Additionally, inhibiting IDH2 in CD8+ T cells and CAR T cells enhances the differentiation of memory T cells, improving the efficacy of cell-based immunotherapies. These findings suggest that wild-type IDH2 is a promising therapeutic target, potentially impacting cancer treatment and immunotherapy by providing new avenues for therapy development. This research underscores the urgent need for specific inhibitors targeting wild-type IDH2, which could revolutionize cancer immunotherapy and broaden treatment options for various cancer types.

**Abstract:**

The metabolic reprogramming characteristic of cancer cells, including the Warburg effect, has long been recognized as a hallmark of malignancy. This commentary explores three recent investigations focusing on the role of wild-type IDH2 in cancer and immune cell function. The first publication identifies wild-type IDH2 as a crucial factor in the survival of triple-negative breast cancer (TNBC) cells, with its inhibition leading to disrupted energy metabolism, reduced tumor growth, and enhanced apoptosis. The second analysis examines the role of IDH2 in CD8+ T cells, revealing that its inhibition promotes the differentiation of memory T cells, thereby enhancing the efficacy of cell-based immunotherapies like CAR T cells. A third investigation supports these findings, demonstrating that IDH2 inhibition in CAR T cells reduces exhaustion, enhances memory T cell formation, and improves anti-tumor efficacy. Collectively, these reports highlight wild-type IDH2 as a promising therapeutic target, with potential applications as a two-edged sword in both cancer treatment and immunotherapy. The development of specific wild-type IDH2 inhibitors could offer new avenues for therapy, particularly in tumors reliant on IDH2 activity as well as in enhancing the effectiveness of CAR T cell therapies.

## 1. Introduction

It has long been known that cancer cells exhibit consistent metabolic changes, notably aerobic glycolysis (the Warburg effect), which is a hallmark feature used in cancer diagnosis [1,2]. In addition, many cancers harbor mutations in genes related to metabolism, including those encoding enzymes such as succinate dehydrogenase (SDH) [3,4], fumarate hydratase (FH) [5], and isocitrate dehydrogenase (IDH) [6]. However, attempts to target cancer metabolism through glycolytic inhibition have yielded mixed results due to the metabolic flexibility of cancer cells.

IDH enzymes catalyze the oxidative decarboxylation of isocitrate, producing α-ketoglutarate (α-KG) and CO_2_. In humans, there are three isoforms of IDH: IDH1, IDH2, and IDH3. IDH1 and IDH2 fuel reversible reactions using nicotinamide adenine dinucleotide phosphate (NADP+) as a cofactor, with IDH1 located in the cytosol and peroxisomes and IDH2 located in the mitochondrial matrix. In contrast, IDH3 catalyzes an irreversible reaction within the tricarboxylic acid (TCA) cycle, generating α-KG and nicotinamide adenine dinucleotide (NADH) [7].

IDH2 normally plays a crucial role in the TCA cycle by converting isocitrate to α-KG in mitochondria. Additionally, IDH2 can participate in the reverse reaction (also known as the reductive TCA cycle or reductive carboxylation) that is a function of the α-KG/citrate ratio and occurs predominantly under hypoxic conditions, producing citrate and acetyl-CoA from glutamine and glutamate [8]. This activity is critical for lipid and cholesterol biosynthesis and is potentially important for cancer cells undergoing active anabolism [9,10]. Beyond their roles in intermediary metabolism and energy production, IDH enzymes are also involved in redox status regulation. Specifically, NADPH maintains an adequate pool of reduced glutathione (GSH), thioredoxin, and catalase tetramers, which are required to counteract the formation of reactive oxygen species (ROS). Additionally, α-KG enables the activity of α-KG-dependent dioxygenase enzymes, which are required for DNA and histone demethylation, DNA repair, HIF degradation, and collagen maturation [7].

While *SDH* and *FH* mutations are considered loss-of-function, IDH1/2 mutant proteins exhibit a new enzymatic activity that catalyzes the NADPH-dependent reduction of α-KG, producing the oncometabolite D-2-hydroxyglutarate (D-2HG) [11]. This oncometabolite blocks cellular differentiation by competitively inhibiting α-KG-dependent dioxygenases involved in histone and DNA demethylation. Additionally, it leads to further alterations in cellular metabolism, redox state, and DNA repair. These discoveries have led to the development of specific IDH1/2-mutant inhibitors, such as ivosidenib [12] and enasidenib [13,14], which are successful examples of targeting cancer metabolism in patients with *IDH1*- or *IDH2*-mutated acute myeloid leukemia. More recently, the dual IDH1/IDH2 inhibitor vorasidenib has been approved for the treatment of *IDH*-mutant glioma following surgery [15].

Interestingly, wild-type IDH2 is highly expressed in various cancers, including glioblastoma, lung, breast, esophageal, colorectal, and hepatocellular carcinomas and has been shown to promote tumor growth and therapy resistance [7,15,16,17,18,19,20,21]. In this commentary, we will analyze three studies emphasizing that wild-type IDH2 could be a potential therapeutic target for specific subsets of breast cancer [22] and that IDH2 inhibition promotes the differentiation of memory CD8+ T cells, thus enhancing the efficacy of cell-based immunotherapies such as chimeric antigen receptor (CAR) T cells [23,24].

## 2. Wild-Type IDH2 Is Essential for Triple-Negative Breast Cancer Cell Survival

Research by Li et al. [22] highlights the identification of wild-type isocitrate dehydrogenase 2 (IDH2) as a potential therapeutic target for a particularly aggressive subset of breast cancer known as triple-negative breast cancer (TNBC). These tumors are so named because they lack estrogen receptor (ER), progesterone receptor (PR), and HER2 expressions and are thus unresponsive to hormonal and HER2-targeted treatments [25]. Consequently, TNBC often results in higher-grade, invasive tumors with limited treatment options and poor clinical outcomes. Through the analysis of genomic datasets, the authors revealed that wild-type *IDH2* mRNA expression was significantly elevated in breast cancer, particularly in TNBC, correlating with advanced clinical stages (Figure 1). Immunohistochemistry confirmed heightened IDH2 protein levels in TNBC tissues, associated with poorer clinical outcomes.

Mechanistically, *IDH2* overexpression promoted TNBC cell proliferation in vitro, evidenced by enhanced colony formation and proliferation assays. Conversely, *IDH2* knockdown suppressed cell growth and induced apoptosis, accompanied by reduced expression of the anti-apoptotic protein MCL-1. In vivo xenograft models further validated IDH2′s essential role in TNBC tumor progression, emphasizing its potential as a therapeutic target.

Contrary to expectations, the authors discovered that *IDH2* knockdown in TNBC cell lines significantly increased cellular α-KG levels both in vitro and in vivo. This elevation was attributed to enhanced glutamine uptake and increased conversion of glutamate to α-KG through phosphoserine aminotransferase 1 (PSAT1), a response linked to metabolic stress and mediated by NRF2 and ATF4 pathways. Metabolic flux analysis using labeled glutamine further confirmed the shift toward α-KG accumulation, highlighting the hindrance of the reductive TCA cycle. Inhibition of IDH2 by AGI-6780 exacerbated α-KG accumulation and suppressed both oxidative and reductive TCA pathways. Rescue experiments reinstating *IDH2* expression reversed α-KG accumulation, underscoring the specificity of IDH2 in modulating α-KG levels and its potential as a therapeutic target in TNBC.

The authors demonstrated that adding cell-permeable α-KG analogs to TNBC cell cultures increased intracellular α-KG levels and significantly inhibited cell growth and induced apoptosis, thus mirroring *IDH2* knockdown. Specifically, knocking down *IDH2* expression as well as boosting α-KG levels disrupted energy metabolism by inhibiting both ATP synthase and glycolysis, resulting in marked ATP depletion.

Furthermore, research shows that *IDH2* expression correlates with HIF-1α signatures and target molecules. Inhibition of *IDH2* accelerates HIF-1α degradation and reduces the expression of its target genes. This can be reversed by stabilizing HIF-1α with prolyl hydroxylase inhibitors or by overexpressing *IDH2*, indicating that *IDH2* regulates HIF-1α protein stability. Moreover, the authors demonstrated that *IDH2* knockdown suppresses TNBC cell invasion, migration, and metastasis in vivo, while overexpression increases the epithelial-to-mesenchymal transition (EMT) and metastatic potential, highlighting the therapeutic potential of targeting IDH2 and HIF-1α pathways in TNBC (Figure 2).

Pharmacological inhibition of wild-type IDH2 using AGI-6780 significantly suppresses the growth of multiple TNBC cells both in vitro and in vivo, mirroring the effects seen with *IDH2* knockdown. Additionally, cell-permeable α-KG also demonstrates significant tumor-suppressing activity, which is further enhanced when combined with the chemotherapeutic drug doxorubicin. This combination likely enhances doxorubicin accumulation in TNBC cells by reducing ATP levels, thus impairing ATP-dependent drug efflux mechanisms.

Overall, Li et al.’s study highlights that wild-type IDH2 is crucial for TNBC cell survival due to its unique metabolic traits, thus suggesting that targeting IDH2 with pharmacological inhibitors like AGI-6780 is a promising therapeutic strategy for TNBC. Future research should expand these findings by identifying other tumors addicted to IDH2 activities [28]. Interestingly, expression profile analysis revealed even higher levels of *IDH2* mRNA in HER2-positive breast cancer compared to TNBC (Figure 1), which warrants further investigation. Moreover, it is essential to precisely dissect the mechanisms through which tumor cells are more sensitive to metabolic and epigenetic disturbances caused by IDH2 inhibition.

## 3. IDH2 Inhibition Promotes the Differentiation of Memory CD8+ T Cells and Enhances the Efficacy of Cell-Based Immunotherapies

The study by Jaccard et al. [24] explores the capacity of effector CD8+ T cells to perform reductive carboxylation of glutamine via the mitochondrial enzyme IDH2, a process crucial for sustaining clonal expansion and antiviral effector functions.

Jaccard’s team observed that highly proliferative effector CD8+ T (T_E_) cells exhibited significantly higher levels of reductive metabolites than memory T (T_M_) cells. This was associated with increased glutamine consumption and fatty acid synthesis. RNA sequencing data showed that *IDH2* had the highest fold-change increase among the IDH isoforms in short-lived effector T cells compared to all memory T cells. *IDH2* deletion in T cells impaired glutamine flow, confirming *IDH2*’s role in mediating reductive carboxylation in CD8+ T cells. Consistently, *IDH2* knock-out increased the proportion of circulating long-lived central memory T (T_CM_) in mice infected with lymphocytic choriomeningitis virus (LCMV)–Armstrong. This was associated with greater re-expansion capacity and prolonged higher expression of memory markers, such as CD62L and TCF1. Furthermore, *IDH2* deletion led to enhanced viral clearance linked with increased production of IL-2, IFNγ, and TNFα.

Then, the authors asked whether this observation could be exploited to enhance the efficacy of cell-based immunotherapies for cancer, such as CAR T cells [29]. Jaccard et al. demonstrated that the treatment with IDH2 inhibitors AGI-6780 and enasidenib (AG-221) during the ex vivo manufacturing of T cells is sufficient to induce the expression of the memory markers CD62L and TCF1 and enrich TM cells. Strikingly, the authors demonstrated that in vitro treatment with the IDH2 inhibitor AGI-6780 enhanced the formation of memory-like CD8+ T cells and the antitumor capacities of CAR T cell therapies in murine melanoma models and human xenograft models of B-cell acute lymphoblastic leukemia and multiple myeloma.

Transcriptional and epigenomic analyses confirmed that IDH2 inhibition induced the enrichment of the memory CD8+ T cell gene signature and increased the accessibility of memory genes such as Sell, Tcf7, and Ccr7. Mechanistically, Jaccard et al. demonstrated that the loss of the reductive TCA cycle after inhibition of IDH2 is compensated by increased glutamine consumption, fatty acid oxidation, and oxygen consumption rate. Their study identifies glutamine as a major anaplerotic carbon source required for the enhanced differentiation of T_M_ cells.

Finally, the study explored how IDH2 inhibition could drive the differentiation of memory T cells through epigenetic mechanisms linked to changes in metabolites. They show that inhibition of IDH2 increases the histone permissive mark H3K4me3, which is associated with memory T-cell differentiation. They demonstrate that TCA cycle metabolites such as succinate, fumarate, and 2-HG compete with α-KG and inhibit histone demethylases like KDM5, thus inducing higher H3K4me3 levels and enhancing T_M_ cell differentiation. In addition, elevated H3K4me3 promotes histone acetylation, a modification that supports the pluripotency of memory T cells. Inhibition of IDH2 leads to increased acetylation, partly driven by fatty acid oxidation (FAO) and citrate-derived acetyl-CoA.

Overall, these findings suggest that IDH2 inhibition can be used during the ex vivo expansion of T cells to promote memory cell differentiation, potentially improving the effectiveness of adoptive cell therapy (ACT) and CAR T cell therapies.

In a second study by Si et al. [23], the researchers screened a library of mitochondria-related compounds to enhance the long-term efficacy of CAR T cells. The IDH2 inhibitor, enasidenib (AG-221) was found to be the most effective in enriching T memory stem cells (T_SCM_) and central memory T cells (T_CM_). Indeed, AG-221-treated CAR T cells showed enhanced memory T cell markers, decreased exhaustion markers, and better survival post co-culture with acute lymphoblastic leukemia cells. Transcriptomic profiling confirmed the enrichment of memory T cell subsets and the reduction in exhausted T cells. Interestingly, the efficacy of *IDH2* inhibition was the same regardless of CAR architecture. AG-221-pretreated CAR T cells reduced tumor burden faster and provided longer remission in mice with established leukemia or solid tumors such as osteosarcoma. However, despite initial eradication, relapse occurred. The authors further demonstrated that in vivo administration of AG-221 prolonged tumor suppression, enhanced CAR T cell persistence in bone marrow, and extended mice survival. To prove the specific action of AGI-221 on IDH2, researchers used shRNA to knock down *IDH2* or *IDH3* in CAR T cells. Like AG-221, *IDH2* silencing significantly diminished IDH2 activity and expression, leading to increased memory T cell subsets, reduced exhaustion, and improved cell viability. On the contrary, *IDH3* knockdown did not impact memory differentiation or exhaustion.

Mechanistically, the authors demonstrated that IDH2 inhibition redirects glucose carbon utilization from glycolysis to the pentose phosphate pathway (PPP). PPP activation increases the production of NADPH and glutathione, crucial for mitigating oxidative stress and enhancing the functionality and viability of CAR T cells. When PPP activity is reduced, these benefits are reversed, demonstrating that the primary advantage of IDH2 inhibition is the reduction in ROS, leading to enhanced CAR T cell performance.

Furthermore, the authors observed that IDH2 inhibition leads to a blockage in oxidative decarboxylation, resulting in a decrease in mitochondrial α-KG and accumulation of citrate and isocitrate. This is followed by increased cytosolic citrate and conversion to acetyl-CoA, a key molecule for histone acetylation. Elevated acetyl-CoA levels enhance H3K27 acetylation, promoting memory T cell differentiation and function.

Histone acetylation changes upon IDH2 inhibition were confirmed by increased H3K27ac levels, specifically at gene promoter regions associated with T cell function and metabolism. This acetylation promotes the transcription of genes involved in memory T cell formation, such as *SELL* and *TCF7*, and enhances IL6R signaling, which supports CAR T cell survival and anti-tumor efficacy. Additionally, IDH2 inhibition leads to a transcriptional shift, reducing glycolysis through increased acetylation of genes involved in metabolic pathways.

In summary, the research by Si et al. suggests that IDH2 inhibits memory CAR T cell formation through epigenetic regulation by controlling cytosolic citrate levels, which in turn influence histone acetylation and gene expression crucial for memory T cell differentiation.

The studies by Jaccard et al. and Si et al. compellingly demonstrate that IDH2 inhibition not only promotes the formation of memory T cells but also enhances the therapeutic efficacy of CAR T cells (Figure 3). However, both investigations have limitations, including an incomplete understanding of metabolic adaptations post-IDH2 inhibition and the dynamics of memory CAR T cells in specific tumor models. Moreover, discrepancies between the two researchers regarding the specific mechanisms by which IDH2 inhibition alters T cell metabolism and epigenetics warrant further investigation. A deeper understanding of these IDH2-dependent metabolic shifts is crucial for optimizing CAR T cell therapy.

## 4. Conclusions

The three studies’ collective findings underscore the significant therapeutic potential of targeting wild-type IDH2 in cancer treatment and immunotherapy. In triple-negative breast cancer (TNBC), wild-type IDH2 has been identified as a crucial driver of tumor survival and proliferation, with its inhibition leading to impaired energy metabolism, reduced tumor growth, and enhanced apoptosis. Furthermore, the inhibition of wild-type IDH2 in CD8+ T cells and CAR T cells promotes the differentiation of memory T cells, thereby enhancing the efficacy of cell-based immunotherapies. These insights suggest that wild-type IDH2 plays a dual role, contributing to both tumorigenesis and immune regulation.

Interestingly, in the absence of specific inhibitors for wild-type IDH2, all three studies described herein utilized small molecules originally designed to target mutant IDH2, such as AGI-6780 and enasidenib (AG-221) (Table 1). These compounds have been shown to inhibit native IDH2, albeit at higher concentrations. These promising findings underscore the urgent need for dedicated efforts to develop specific inhibitors for wild-type IDH2 [30]. Furthermore, nanovectors could reveal a compelling strategy for the precise delivery of IDH2 inhibitors to target cells [31].

The implications of these studies extend beyond TNBC, potentially offering new therapeutic avenues for a broader range of cancers reliant on IDH2 activity, both as standalone therapies and in combination with other treatments [32,33]. Additionally, enhancing the effectiveness of CAR T cell therapies through IDH2 inhibition could revolutionize cancer immunotherapy. Future research should focus on refining IDH2 inhibitors, exploring their use in various cancer types, and fully elucidating the mechanisms underlying IDH2’s role in both oncogenesis and immune cell function. Potential pathways of resistance to IDH2 inhibitors will also need to be investigated.
cancers-16-03280-t001_Table 1Table 1IDH2 inhibitors. The biochemical potency of IDH2 inhibitors, expressed as inhibitory activity (IC50 value ± SD) against wild-type (WT) or mutated IDH2 isoforms, was evaluated by measuring the conversion of isocitrate to α-KG (for WT IDH2) or α-KG to 2HG (for mutated IDH2) using a fluorescent assay of NADP+/NADPH levels after 1 h of incubation (except for C6, where 15 min incubation was reported). The standard deviation (SD) is provided when available, with “N/A” indicating data not available. Chemical structures were designed using ChemDraw 18.1.IDH2 Inhibitor StructureReported TargetIDH2 (WT) IDH2 R140QIDH2 R172QIDH1 (WT)IDH1 R132HCompanyDevelopment Phase Ref.IC50 (uM)IC50 (uM)IC50 (uM)IC50 (uM)IC50 (uM)**AG-221 (Enasidenib)**
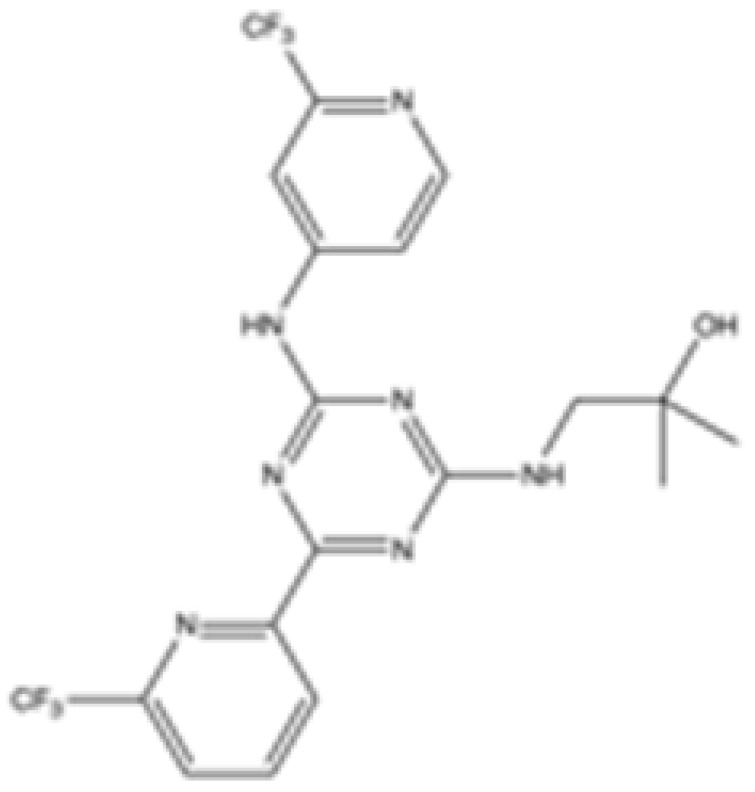
**mIDH2**39.83 ± 9.080.32 ± 0.050.2 ± 0.071.12 ± 0.6877.64 ± 11.92AgiosApproved[13]**7c**
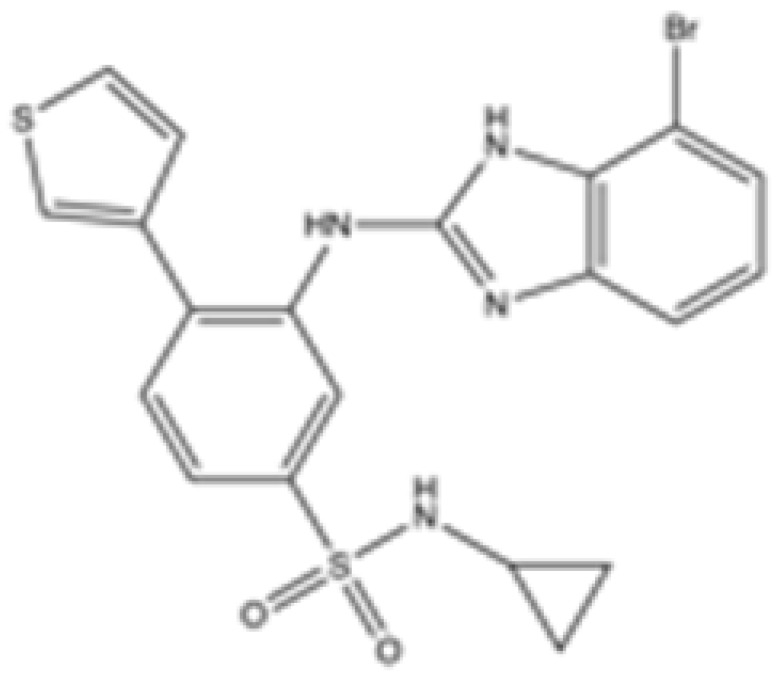
**mIDH2**>1000.26 ± 0.04>100>100>100
Preclinical[30]**AGI-6780**
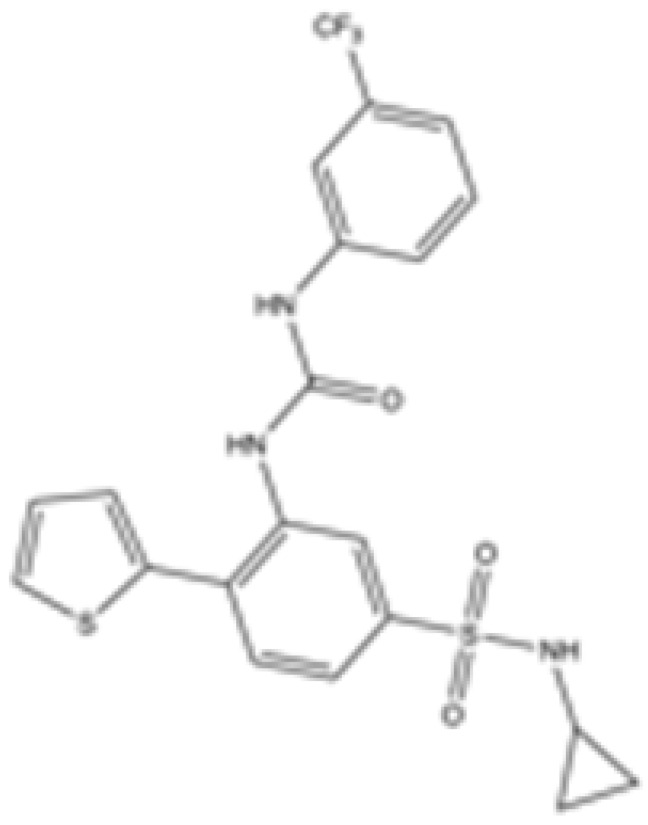
**mIDH2**0.27 ± 0.0310.017 ± 0.047N/AN/AN/AAgiosPreclinical[34]**IDH2-C100**
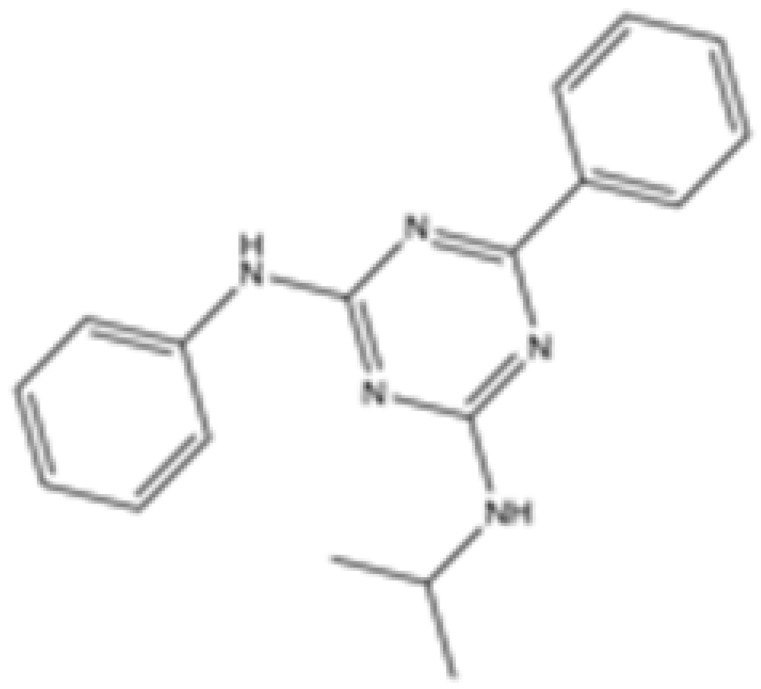
**mIDH2**N/A<0.1<0.1N/AN/AAgiosPreclinical [35]**AG-881 (Vorasidenib)**
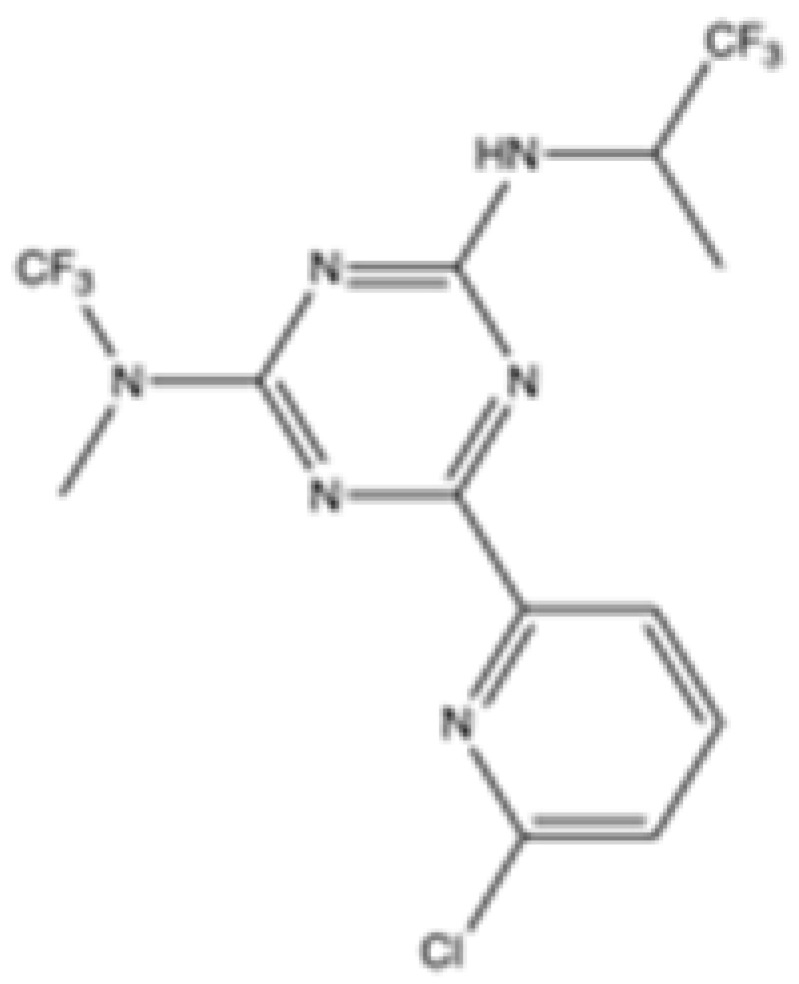
**mIDH1/IDH2**0.374 ± 0.0360.118 ± 0.0140.032 ± 0.0040.190.006 ± 0.002AgiosApproved[36]**HMPL-306**
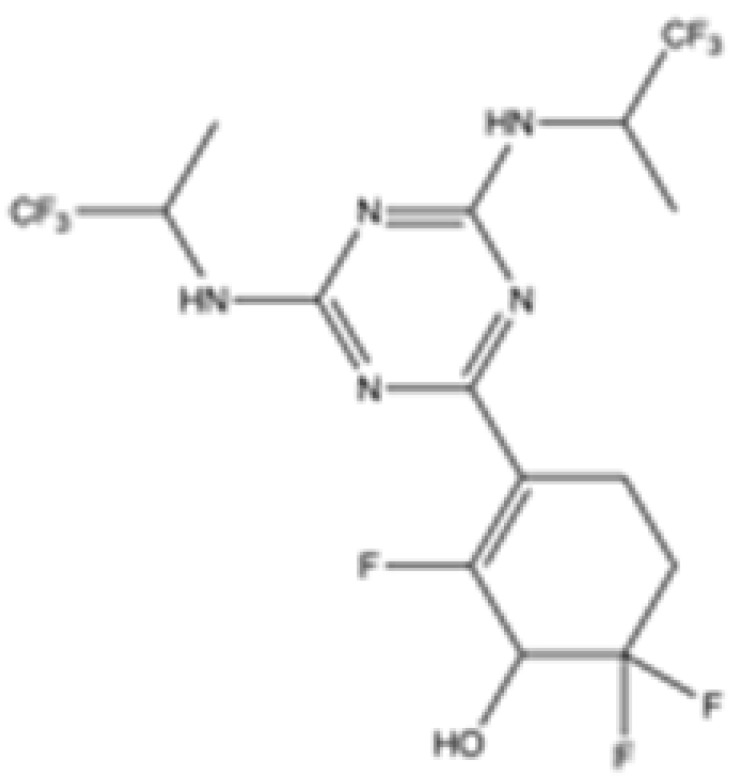
**mIDH1/mIDH2**N/AN/AN/AN/AN/AHutchmedClinical trial phase III[37]**C6**
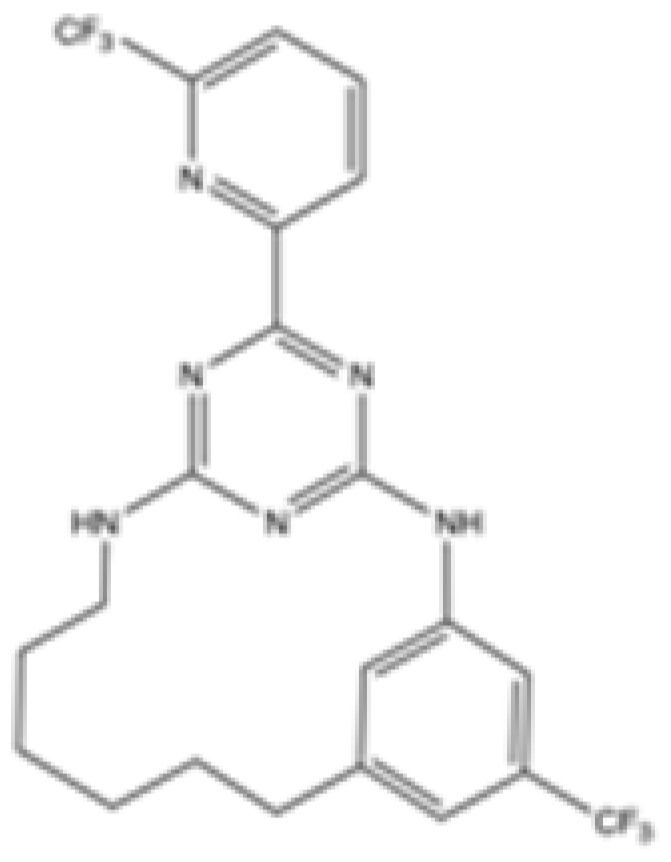
**mIDH2**>20006.1 ± 1.9218.9 ± 9.4>20006.1 ± 1.9
Preclinical[38]**SH-1573**
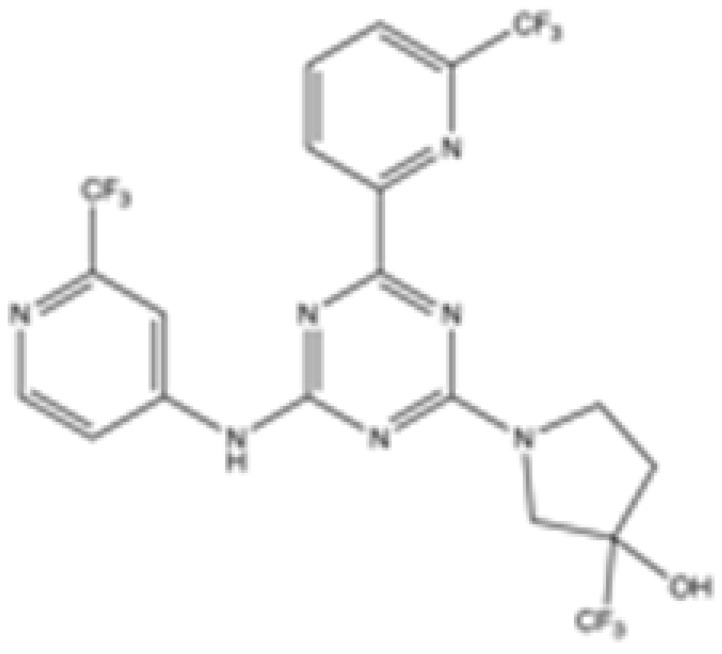
**mIDH2**0.19620.00480.0141>100>100
Clinical trial phase I[39]**Abrine**
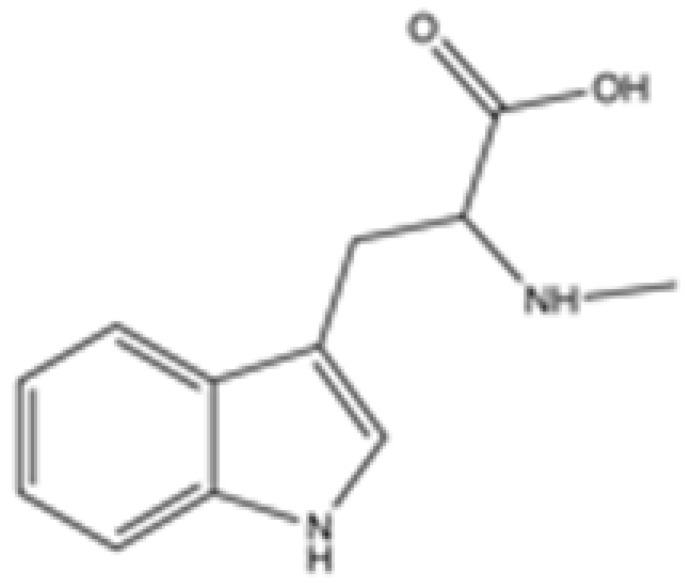
**mIDH2**N/AN/AN/AN/AN/A
Preclinical[40]**Precatorine**
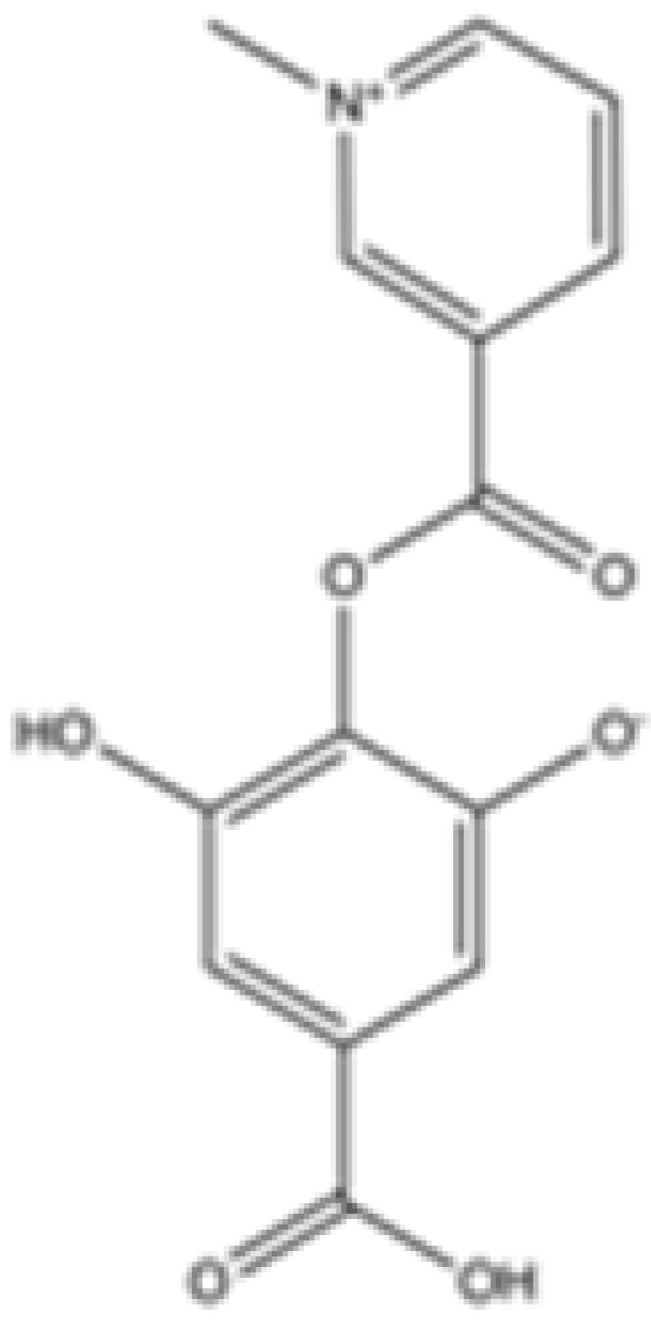
**mIDH2**N/AN/AN/AN/AN/A
Preclinical[40]**Compound 6b**
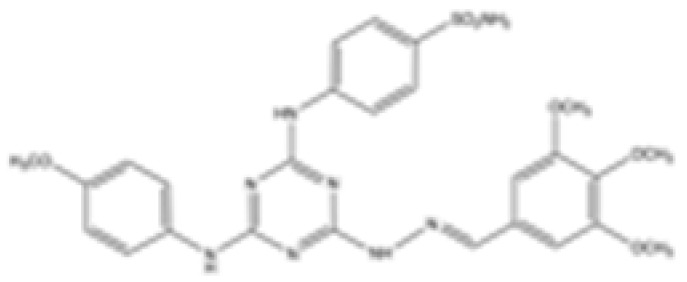
**mIDH1/mIDH2**N/A1.6 ± 0.032N/AN/A0.22 ± 0.005
Preclinical[41]**36**
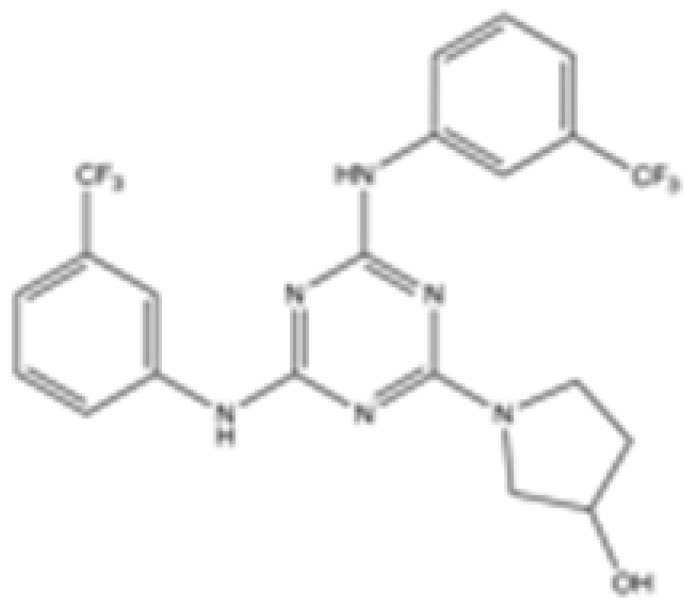
**mIDH2**>1000.029 ± 0.00020.24 ± 0.014N/AN/A
Preclinical[42]


## Figures and Tables

**Figure 1 cancers-16-03280-f001:**
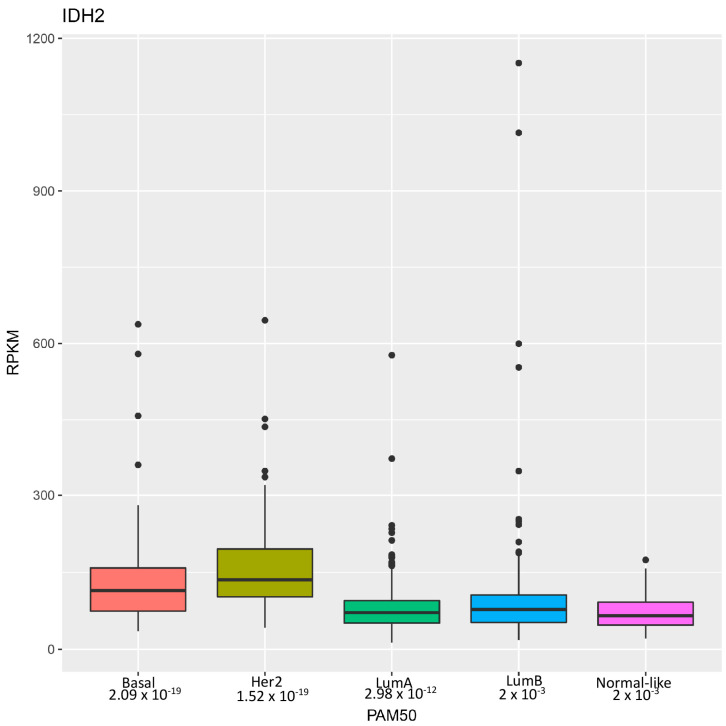
Distribution of *IDH2* mRNA levels in breast cancer subtypes. Distribution of *IDH2* expression levels across different breast cancer subtypes in a breast invasive carcinoma dataset from The Cancer Genome Atlas [26] analyzed using the PAM50 classifier as previously described [27]. The *x*-axis represents the breast cancer subtypes, including luminal A, luminal B, HER2-enriched, basal, and normal-like. The *y*-axis indicates the expression levels of *IDH2*, measured in RPKM (Reads Per Kilobase of transcript per Million mapped reads). Each box plot displays the median, interquartile range, whiskers and outliers of *IDH2* expression within each subtype.

**Figure 2 cancers-16-03280-f002:**
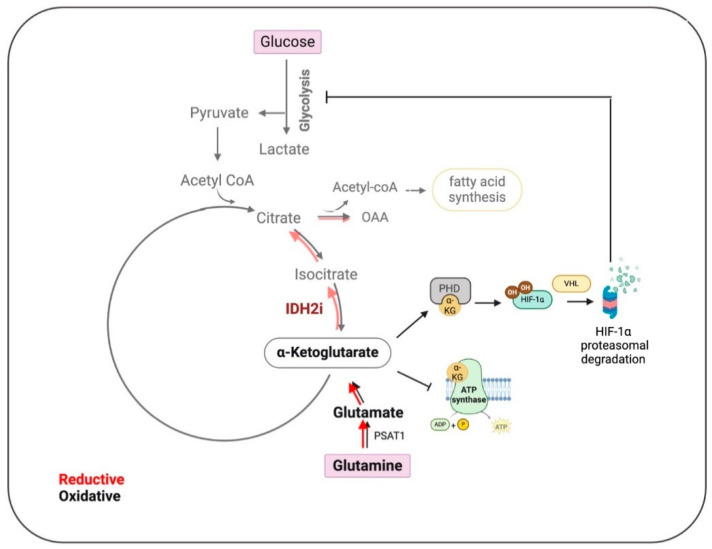
Impact of IDH2 inhibition on the metabolic pathways in Triple-Negative Breast Cancer (TNBC). In TNBC, IDH2 mainly controls the reductive tricarboxylic acid (TCA) cycle, which is slowed down by inhibition of IDH2 (IDH2i) (lighter shade red arrows), leading to the accumulation of alpha-ketoglutarate (*α*-KG). This accumulation promotes the degradation of hypoxia-inducible factor 1 (HIF1), thereby impeding glycolysis, lipid synthesis, and ATP production. These metabolic disruptions collectively result in reduced tumor cell proliferation, inhibition of epithelial–mesenchymal transition (EMT), and decreased metastasis. Oxalacetate (OAA), phosphoserine aminotransferase 1 (PSAT1), prolyl hydroxylase (PHD), and E3 ubiquitin ligase von Hippel–Lindau (VHL). (Created using BioRender.com accessed on 21 August 2024).

**Figure 3 cancers-16-03280-f003:**
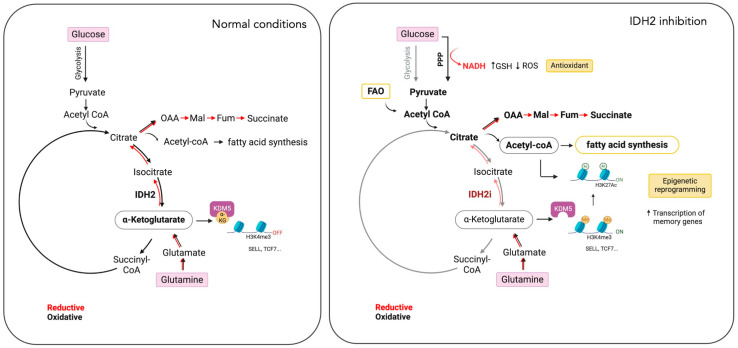
Impact of IDH2 inhibition on T cell metabolism and differentiation. IDH2 inhibition (IDH2i) in T cells disrupts both the oxidative and reductive TCA cycles (lighter shade red and black arrows). In response, T cells increase glutamine uptake, fatty acid oxidation (FAO), and the pentose phosphate pathway (PPP). These metabolic adaptations support lipid synthesis, reduce reactive oxygen species (ROS) levels, and induce epigenetic reprogramming, leading to the differentiation of T cells into memory cells. This metabolic strategy presents a potential approach to enhance the efficacy of CAR T cell therapies. Oxalacetate (OAA), malate (Mal), fumarate (Fum), alpha-ketoglutarate (*α*-KG), selectin L (SELL), transcription factor-7 (TCF7), and lysine demethylase 5A (KDM5A). (Created by BioRender.com accessed on 21 August 2024).

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
