# Peer review of "IDH2 Inhibitors Gain a Wildcard Status in the Cancer Therapeutics Competition"

_cancers, 2024, doi:10.3390/cancers16193280_

Round 1

Reviewer 1 Report

Comments and Suggestions for Authors

The present commentary, “IDH2 Inhibitors Gain a Wildcard Status in the Cancer Therapeutics Competition,” comprehensively explores three recently published articles on IDH2 and its role in cancer and immune response. With its thorough analysis, this commentary is very well written and should be accepted in its present form by Cancers. Below are specific comments:

1.      The authors have provided an in-depth introduction and review of isocitrate dehydrogenase (IDH) and its isoforms, functions, and implications in cancer. They have also provided additional information about IDH2 and its gain-of-function mutations implicated in cancer progression.

2.      The first article discussed in this commentary is by Li et al. 2024, in which Wild-Type IDH2 is presented as a therapeutic target for Triple-Negative Breast Cancer (TNBCs). The commentary summarizes this study and underscores its promising future implications, offering hope for the potential of IDH2 inhibitors in other breast cancer types.

3.      The second article discussed in the commentary is by Jaccard et al. 2023, which explores the reductive carboxylation of glutamine via the mitochondrial enzyme IDH2 by effector CD8+ T cells, which is essential for sustaining clonal expansion and antiviral effector functions. The current commentary discussed the whole article and its results comprehensively, providing details of significant findings.

4.      The third article discussed is by Si et al. 2024, wherein IDH2 inhibition was the most effective in enriching T memory stem cells (TSCM) and central memory T cells (TCM). Through the commentary, the authors provide a detailed overview of the synergic complementation of Si et al. 2024 and Jaccard et al. 2023 on IDH2 inhibition to promote the formation of memory T cells and improve CAR T therapy efficiency.

5.      The commentary discussed the limitations of each reviewed article. Additionally, the concluding remarks provide an essential outlook on the potential future directions for developing IDH2 as a therapeutic target in cancer and improving CAR-T immunotherapies. This insight could inspire further research and innovation in cancer therapeutics. 

Author Response

Thank you for your thorough and positive review. We appreciate your insightful comments and recognition of the comprehensive nature of our commentary.

Reviewer 2 Report

Comments and Suggestions for Authors

Comments to Roberto Piva et al., IDH2 inhibitors,

The authors provided a useful piece of commentary focusing on the expected clinical benefits of inhibitors that target wild type IDH2. They picked up three papers Li et al., Nature Comm. 2024, Si et al., Cell Metab. 2024 and Jaccard et al., Nature 2023. By analyzing these papers, they demonstrated that IDH2 could be a driver in TNBC and that treatment with IDH2 inhibitor may be therapeutically benefitable to patients by promoting differentiation of CD8+ T cells and CAR-T cells. The correctly understood these works and properly introduced their take home lessons to expected readers. Only two wild type IDH2 inhibitors (AGI-6780 and AG-221) appeared in the text. If they can provide a Table on more detailed information on these chemicals (structure, IC50, specificity, toxicity, refs, etc.) and introduce as many as possible other currently available IDH2 inhibitors, this commentary would be more informative.  

Author Response

Thank you for your valuable feedback and suggestions. We appreciate your recognition of the commentary’s relevance and its potential clinical implications regarding IDH2 inhibitors. In response to your request, we have incorporated a table with detailed information on IDH2 inhibitors, including their structure, IC50 values, specificity, and references. This addition will provide readers with a more comprehensive overview of available inhibitors.